# Losses for Deep Probabilistic Regression

## Abstract

Probabilistic regression is used in fields such as healthcare, finance, energy, robotics and meteorology. Although many works have dealt with probabilistic regression, they have frequently done so independently, often failing to compare against each other. This paper reviews probabilistic regression and aims at providing a unified overview of the area. We experimentally compare diverse approaches and observe that direct methods perform comparably to their sample-predicting counterparts, while being simpler to train and cheaper to infer with. We then introduce a taxonomy that sheds light onto the design choices behind each of the direct methods, suggesting new ones. The main takeaway is that simple methods can serve as strong baselines and should not be disregarded.

## 1. Introduction

Probabilistic regression is a field shared by various disciplines, such as finance (Timmermann, 2000), meteorology (Ravuri et al., 2021; Bi et al., 2023; Wilks, 2011), statistics (Gneiting & Katzfuss, 2014) and machine learning (Danelljan et al., 2020; Bishop & Nasrabadi, 2006). Some of these disciplines developed probabilistic regression independently and out of necessity, as they require reliable estimations of the probabilities of all outcomes. Probabilistic predictions (including forecasting) are also encountered in healthcare (Jones & Spiegelhalter, 2012; Alkema et al., 2007), energy (Zhang et al., 2014; Xu et al., 2022; Lauret et al., 2024), hydrology (Krzysztofowicz, 2001), economics (Timmermann, 2000), demographics (Raftery et al., 2012) and computer vision (Gustafsson et al., 2020b) applications, among others. As a consequence, works on probabilistic regression are scattered throughout the scientific literature, many times unaware of each other (e.g. (Han et al., 2022) and (Gustafsson et al., 2022)). Furthermore, there is no

consensus on baselines and different communities use different metrics. The aim of this review paper is to provide an entry point for practitioners of deep probabilistic regression, i.e., of probabilistic regression that leverages the powerful representations obtained with deep networks.

The research question guiding our work is: *"What is the best probabilistic regression method?"*. Naturally, goodness criteria should be defined, and the usual considerations of performance, efficiency, ease of use, and scalability are obviously of interest. Methods that mirror supervised learning, except for the chosen loss function, are particularly attractive when considering efficiency, ease of use and scalability. We call these methods *direct* methods. Unfortunately, there are many direct methods, presented in individual disconnected papers, and largely ignored in past surveys on probabilistic predictions. For readers already familiar to probabilistic regression, **the main contributions of this work are (a) the collection and categorization of direct methods under a unifying taxonomy and (b) the experimental comparison against non direct methods.**

We have also found that the literature contains many different yet related concepts (e.g. probabilistic regression, forecasting, uncertainty estimation, calibration) and many different metrics and evaluation tools. For instance, meteorology and epidemiology make heavy use of the Continuous Ranked Probability Score (CRPS) as a metric (Wilks, 2011; Bracher et al., 2021), while in machine learning, the negative log-likelihood (NLL) is more commonly used (Bishop & Nasrabadi, 2006). For readers new to probabilistic regression, we **(c) provide an entry-point describing the main concepts and standard evaluation practices** in Section 3.

In a nutshell, this work provides a new taxonomy, summarized in Table 1, that organizes and helps us understand direct methods, which are cheap, scalable, and easy to use. The taxonomy suggests new methods and avenues for improvement. It involves framing deep probabilistic regression similarly to supervised learning, looking at the target loss (either NLL or CRPS, presented later), at the fixed, learned or predicted parameters, and at the Cumulative Distribution Function (CDF) they assume. We also provide a brief introduction to probabilistic regression, including strictly proper scoring rules, calibration considerations, and categorize methods into Bayesian, ensemble, generative, and

---

[1]Anonymous Institution, Anonymous City, Anonymous Region, Anonymous Country. Correspondence to: Anonymous Author <anon.email@domain.com>.

Preliminary work. Under review by the International Conference on Machine Learning (ICML). Do not distribute.

direct. Approximations and differentiable formulations of the CRPS are also reintroduced. Besides these conceptual contributions, we run an extensive (8 datasets, 20 fold cross-validation) apples-to-apples comparison between sampling methods and direct methods. We show that direct methods are comparable to their sample-predicting counterparts. This work aims to structure the field and provide strong, lightweight baselines that accelerate the development of effective deep probabilistic regressors across disciplines.

The paper is organized as follows: Section 2 introduces **related work**, with a focus on ensemble, Bayesian, generative and direct methods. Section 3 provides **background** on probabilistic regression and its evaluation via scoring rules. Section 4 introduces and explains the most representative **direct methods** found in the literature. The **taxonomy** presented in Section 5 organizes direct methods according to their design choices and suggests new losses. The most representative direct methods are compared to sample-predicting approaches in the **experiments** of Section 6. Finally, Section 7 **concludes** the work.

## 2. Related Work

Many works have reviewed probabilistic prediction methods in the past, often restricted to specific disciplines, for instance meteorology (Wilks, 2011), epidemiology (Bracher et al., 2021), or wind forecasting (Bazionis & Georgilakis, 2021). Some reviews have dealt with probabilistic predictions in general, notably (Gneiting & Katzfuss, 2014). More related to our work are reviews of probabilistic predictions in a deep learning context (Tyralis & Papacharalampous, 2024; Abdar et al., 2021; Seligmann et al., 2024). Unfortunately, they focus mostly on classification, ensembles and Bayesian inference, and do not pay much attention to what we call *direct* methods for regression, which are the focus of this work.

Probabilistic regression is related to uncertainty estimation. In fact, probabilistic regression is evaluated using strictly proper scoring rules (SPSRs), similar to uncertainty estimation (Gustafsson et al., 2020b). The main difference is in intent: while probabilistic regression aims to provide probability estimates for every outcome, uncertainty estimation is about detecting unreliable predictions. Probabilistic regression is also related to calibration (Dheur & Taieb, 2023; Minderer et al., 2021). Indeed, SPSRs automatically take calibration into account, as they measure how close the predicted distribution is to the observed data distribution. All else being equal, uncalibrated models get worse scores than calibrated ones.

**Ensemble methods** have been used for a long time in meteorology (Richardson, 2000). They generate a probability distribution from different point predictions, usually by com-

puting the empirical CDF. Ensembles implicitly assume that the predictions are distributed similarly to the target variable. However, this is not necessarily the case, for example, in numerical weather prediction, small physically plausible perturbations of the initial conditions (Anderson, 1997) are not always mapped through the simulation to the true distribution of the outcomes, and statistical postprocessing is frequently required (Gneiting & Katzfuss, 2014; Wilks, 2011). The principle of adding perturbations to the input has been applied also in deep learning (Wen et al., 2020). Ensembles of deep models (a.k.a. DeepEnsembles) were originally introduced in (Lakshminarayanan et al., 2017). Albeit old, DeepEnsembles perform the best when scaled up (Gustafsson et al., 2020b; Seligmann et al., 2024). It is worth noting that DeepEnsembles are simply a collection of many direct methods, therefore all advances in direct methods translate easily into ensembles.

**Bayesian Methods** model the distribution of the target variable $y$ conditioned on an input $x$ by marginalizing the parameters $w$. In other words, Bayesian methods aim to use the distribution $p(w|D)$ of the model parameters $w$ given the data $D$. They compute

$$p(y|x, D) = \int p(y|x, w)p(w|D)dw, \qquad (1)$$

with the integral being approximated by Monte Carlo sampling of $w_i \sim p(w|D)$. However, sampling from $p(w|D)$ is unfeasible, and approximations are needed (Gustafsson et al., 2020b). Equation 1 is known as the Bayesian Model Average (BMA) (Wilson, 2020). It is worth noting that DeepEnsembles can be seen as approximation of the BMA (Wilson, 2020; Gustafsson et al., 2020b). A celebrated example of Bayesian inference is Monte Carlo Dropout (MCD) (Gal & Ghahramani, 2016) (refined in (Hron et al., 2018)), which uses dropout at training and inference time. Bayesian networks (Pearl, 2022) were also foundational, although it is hard to make some new architectures bayesian (Cinquin, 2021). Probabilistic Backpropagation (PBP) is also of interest (Hernández-Lobato & Adams, 2015). We experimentally evaluated PBP and MCD which resulted in inferior performance relative to most other approaches. On top of that, Bayesian methods usually involve sampling, which adds computational overhead at inference and is harder to implement (Seligmann et al., 2024).

**Generative models** are related to ensembles and Bayesian methods in that they also generate samples of the target distribution. Some generative models are conditioned on a latent variable $\epsilon$, which is sampled at random during training and inference (Ravuri et al., 2021; Zhao et al., 2016). Other two relevant works are GCDS (Zhou et al., 2023), an extension of the GAN (Zhao et al., 2016), and CARD (Han et al., 2022), which introduces diffusion for probabilistic regression. Notably, CARD outperforms DeepEnsembles.

*Table 1.* Example characterization of representative direct methods for probabilistic regression under our taxonomy.

| Name | Reference | Minimizes | Implicit | Predicts | CDF |
|------|-----------|-----------|----------|----------|-----|
| **Canonical** | (Bishop, 1994; Nix & Weigend, 1994) | NLL | ✗ | $(\mu, b)$ or $(\pi_k, \mu_k, \sigma_k)_{k=1}^K$ | Laplace, Normal, MoG |
| **Canonical** | (Dheur & Taieb, 2023) | CRPS | ✗ | $(\pi_k, \mu_k, \sigma_k)_{k=1}^K$ | Normal |
| **CE** | (Stewart et al., 2023) | NLL | ✗ | $\{\tau_i = \text{CDF}(b_i)\}_{i=0}^B$ | Piecewise-Linear |
| **Pinball** | (Koenker & Bassett, 1978) | ≈CRPS | ✗ | $\{b_i : \text{CDF}(b_i) = \tau_i\}_{i=1}^{B-1}$ | Piecewise-Linear |
| **IQN** | (Dabney et al., 2018a) | ≈CRPS | ✓ | $b_\tau$ for any $\tau$ | $F(y) = \sup\{\tau : b_\tau \leq y\}$ |
| **EBM** | (Gustafsson et al., 2020a; 2022) | NLL | ✓ | $f(y)$ for any $y$ | $F(y) = \int_{-\infty}^y f(y')dy'$ |

**Direct methods** do probabilistic prediction more simply, mainly leveraging a loss function. Throughout the paper, we describe the contributions of Hamilton et al.; Bishop & Nasrabadi; Bishop; Nix & Weigend regarding NLL minimization. We also experiment with regression-by-classification approaches (Sønderby et al., 2020; Oord et al., 2016), and show it makes theoretical sense as it is simply another instance of proper scoring rule minimization. The Pinball loss might be tracked back to (Koenker & Bassett, 1978) and it has been present in the literature ever since (Steinwart & Christmann, 2011) and even improved upon (Chung et al., 2021). To the best of our knowledge, the CRPS as a deep learning loss was used in (Dheur & Taieb, 2023) for the Mixture Density Network (MDN) (Bishop, 1994) but never for piecewise-linear CDFs. The implicit models of Implicit Quantile Network (IQN) (Dabney et al., 2018a) and its density-estimation analog, Energy Based Model (EBM) (Gustafsson et al., 2020a), is also included in our review.

## 3. Background

We describe probabilistic regression in terms of an input to the regressor $x \in \mathbb{R}^d$, a continuous outcome variable $y \in \mathbb{R}$, and a predicted cumulative density function (CDF) $F(y|x) = P(Y \leq y|x)$ or its derivative the probability density function (PDF) $f(y|x)$ (both conditioned on $x$). Unlike deterministic regression, which outputs a single point estimate $\hat{y} = f(x)$, probabilistic regression seeks to predict the true conditional distribution $p(y|x)$ (Gustafsson et al., 2020a). For problems with many variables (e.g. different locations in weather forecasting (Sønderby et al., 2020)), one can predict multiple distributions $(p(y_1|x), \ldots, p(y_N|x))$ for some input data $x$. This is *different* from modeling the joint distribution $p(y_1, \ldots, y_N|x)$, which is seldom required and outside the scope of our work (see (Gustafsson et al., 2020a) if multidimensional distributions are needed). Obviously, one has no access to $p(y|x)$, only to samples $\{(x_1, y_1), \ldots, (x_N, y_N)\}$, which one can use to evaluate how well a probabilistic regression method performs.

Fortunately, there is consensus on how to evaluate proba-

bilistic predictions. While other metrics can be proposed, the ones below provide a rather complete overview of how probabilistic regression is evaluated in diverse scientific disciplines, and are applicable to any method. Some metrics are classified as *strictly proper scoring rules* (SPSRs), meaning that they are only minimized when the predicted distribution exactly matches the true distribution of the data (Gneiting & Raftery, 2007). It follows that the optimal solution is the same across these metrics (the true distribution), but they provide different numerical values and optimization landscapes. We comment on categorical distributions in the Supplement D, and focus now on the methodology for probabilistic regression tasks. The two most widely used strictly proper scoring rules for regression are the Continuous Ranked Probability Score (CRPS) and the LogScore (LS), also known as negative-log-likelihood (NLL) (Gneiting & Katzfuss, 2014). Although they are known as "scores", *lower is better*. In what follows, we drop the conditioning on the input $x$ for the sake of brevity and generality.

The **CRPS** is defined as

$$\text{CRPS}(F, y) = \int_{-\infty}^{\infty} (F(y') - \mathbf{1}_{y \leq y'})^2 dy'. \quad (2)$$

The CRPS evaluates the entire predicted CDF, $F(y)$, and compares it with a step function centered at the observed value. It is a strictly proper scoring rule (Matheson & Winkler, 1976). Sometimes, it is the only metric used to evaluate probabilistic regression performance (Ravuri et al., 2021). The most general way to compute the CRPS is numerically, but for specific distributions such as the ones given by piecewise-linear CDFs, closed-form formulas can be derived, which are exact and fast (Suplement F). The CRPS is robust to estimation errors, and this makes it preferable in some applications (Bracher et al., 2021).

The **LogScore** (LS) or **Negative Log Likelihood** (NLL) is defined as

$$\text{LogScore}(f, y) = \text{NLL}(f, y) = -\log(f(y)). \quad (3)$$

This involves evaluating the predicted PDF $f$ at the observed value $y$. Like the CRPS, it is strictly proper, but unlike the

CRPS, it presents the property of *locality*, i.e. the metric does not depend in probabilities assigned to values other than the observed (which means that bin contiguity is ignored for histogram-like PDFs). It is also additive, meaning that the LogScore of many predictions is the sum of the individual LogScores (Benedetti, 2010). Moreover, the LS is useful for classification tasks and is trivially differentiable. The expected LS is equivalent to the Kullback–Leibler divergence $D_{\mathrm{KL}}(p\|f)$ up to the constant entropy of the true distribution $\mathbb{E}_{p(y)}[-\log p(y)]$. The NLL diverges to infinity if any observed outcome was assigned a zero probability, which might or might not be desirable, depending on the application.

**Sharpness and calibration** offer a holistic view of a model's performance. Sharpness refers to the concentration of predictive distributions and depends only on the predictions. Higher sharpness is better, given that the model is calibrated. Calibration refers to the alignment between predicted probabilities and observed frequencies. A probabilistic regression model is calibrated when the frequency of the actual outcomes falling within a specified predictive interval or quantile corresponds with the predicted frequency. We define $\tau_i = F(y_i|x_i) = P(Y \leq y_i|x_i)$. Computing $\tau_i$ for each data pair $(x_i, y_i)$ yields the collection of pairs $(y_i, \tau_i)$. These pairs serve as the basis for calculating calibration metrics and generating insightful plots, as discussed below.

A **reliability diagram** offers a visual method to assess the calibration of probabilistic predictions. It leverages the Probability Integral Transform (PIT) theorem (Dodge, 2003), which states that $\tau = F(Y)$ should be uniformly distributed, therefore their empirical CDF should be the identity function. The reliability diagram is defined as the scatter plot of the pairs $(\tau_{(i)}, i/n)$ where $\tau_{(1)} < \tau_{(2)} < \cdots < \tau_{(n)}$ is the ranking of the $\tau$s. Compared to the PIT histogram, the reliability diagram looks less noisy.

Lastly, the **Expected Calibration Error** (ECE) provides a quantitative measure of a model's calibration by computing the mean absolute deviation between the predicted cumulative probabilities and the observed frequencies. For cumulative probability predictions $\tau_i$, the calibration error is calculated as

$$\mathrm{ECE} = \frac{1}{n} \sum_i \left| \tau_i - \frac{1}{n} \sum_j 1_{\tau_j \leq \tau_i} \right|, \qquad (4)$$

where $n$ is the total number of observations. Graphically, it is the mean absolute difference between the Reliability Diagram curve and the identity function. The ECE offers a concise metric that summarizes the overall calibration of the model. However, it does not indicate with what probability or how the model is miscalibrated.

# 4. Review of Differentiable Losses for Probabilistic Regression

We now describe and present the menu of direct methods using consistent vocabulary and notation, highlighting similarities and ordering them by incremental complexity.

**Direct methods for deep probabilistic regression** mirror traditional deep regression. We denote a neural network with weights $w$ and input $x$ by $g$ and call its outputs $z = g(x, w)$. Training a neural network for traditional regression involves solving

$$w^\star = \arg\min_w \frac{1}{N} \sum_i L(g(x_i, w), y_i), \qquad (5)$$

where $(x_i, y_i)$ are input-target pairs and $L$ is a loss function such as the Mean Squared Error (MSE) or the Mean Absolute Error (MAE). Analogously, **direct** probabilistic regression methods solve

$$w^\star = \arg\min_w \frac{1}{N} \sum_i \mathrm{SPSR}(f_{g(x,w)}, y_i), \qquad (6)$$

where $f_z = f_{g(x,w)}$ is a probability distribution explicitly parametrized by[1] $z = g(x, w)$ and SPSR is a strictly proper scoring rule. Here, $z$ is a set of parameters that describe the distribution, different models will yield different $z$. In this case, the SPSR serves the role of optimization objective, not of evaluation metric (if desired, it can be used as such too). Equation 6 can be rewritten to consider a loss function $L_f(z, y) = \mathrm{SPSR}(f_z, y)$ that acts on the parameters $z$ and not on the distribution $f_z$:

$$w^\star = \arg\min_w \frac{1}{N} \sum_i L_f(g(x, w), y_i), \qquad (7)$$

exactly matching traditional regression. The attractiveness of direct methods is that most advances in traditional supervised learning translate directly to probabilistic regression.

## 4.1. Deterministic Predictions Seen as Probabilistic

Equations 5 and 7 are the same up to the choice of the loss function, where $L_f$ is related to some probability distribution $f$. But in fact, for the cases of the MSE and MAE, the equations are *exactly* the same. Let $f$ be a Gaussian $\mathcal{N}(\mu, \sigma)$, formally

$$p(y|x) = \frac{1}{\sqrt{2\pi\sigma^2}} \exp\left(\frac{-1}{2\sigma^2} \|y - \mu\|^2\right). \qquad (8)$$

Its LogScore or NLL is

$$-\log p(y|x) = \frac{1}{2}\left(\log 2\pi\sigma^2 + \frac{\|y - \mu\|^2}{\sigma^2}\right) \qquad (9)$$

---

[1]We will abuse notation slightly: sometimes $g$ will be followed by some manipulation (e.g., an activation function) before obtaining $z$.

which is to be minimized (Equation 7). For predicted $\mu = g(x, w)$ and fixed $\sigma = \sqrt{2}^{-1}$ the equation becomes

$$- \log p(y|x) = \frac{\log \pi}{2} + \|y - g(x, w)\|^2, \qquad (10)$$

which is exactly the MSE up to an additive constant that does not modify the optimization. Therefore, for this choice of fixed $(\sigma)$ and predicted $(\mu)$ parameters, minimizing the LogScore ($L_f = -\log p(y|x)$) amounts to minimizing the MSE. The same happens for the MAE, but with the Laplace distribution:

$$L_f(z, y) = \log(2b) + b^{-1}|y - \mu|, \qquad (11)$$

where $z = \mu = g(x, w)$ and $b = 1$. The equivalence between the optima (which are the mean and median of the true $p(y|x)$ for the MSE and MAE, respectively) happens irrespective of the choice of the distribution width ($\sigma$ or $b$). This well-known fact is explained in (Bishop & Nasrabadi, 2006) and justifies the MSE and the MAE.

### 4.2. Learned Parameters

It is interesting to note that the choice of which parameter to predict ($\mu$) and which parameter to fix (the width) is arbitrary, one could have chosen to predict the width and fix the $\mu$ instead. In fact, a third way to handle parameters was introduced in (Hamilton et al., 2020) which involves *learning* a global parameter. In particular, they showed that learning the "fixed" width $b$ or $\sigma$ of the assumed distribution during training was useful for outlier detection, robust modeling, and recalibration. This approach is a step closer to what is commonly known as probabilistic regression.

### 4.3. Full Likelihood Maximization

One of the key concepts in (Bishop & Nasrabadi, 2006) is log-likelihood maximization. This is equivalent to minimizing the LogScore. LogScore minimization is applicable to any distribution $f$ (as long at it does not diverge). One can choose to predict *all* the parameters of the distribution $f$. For the Laplace case, this entails to predicting $(\mu, b) = (g(x, w)_1, \text{Softplus}(g(x, w)_2))$, where the second variable should be positive. For the Gaussian case, we can similarly set $(\mu, \sigma) = (g(x, w)_1, \text{Softplus}(g(x, w)_2))$. The Gaussian case is exactly the Mixture of Gaussians (MoG) distribution for $K = 1$, and in this case the loss becomes

$$L_f(z, y) = \log \left( \sum_k \pi_k \frac{1}{\sqrt{2\pi\sigma_k^2}} \exp \left( \frac{-1}{2\sigma_k^2} \|y - \mu_k\|^2 \right) \right), \qquad (12)$$

with $z = \{(\pi_i, \mu_i, \sigma_i)\}_i$, mixture weights $(\pi_1, \ldots, \pi_K) = \text{Softmax}(g(x, w)_1, \ldots, g(x, w)_K)$, centers $\mu_i = g(x, w)_{K+i}$, and widths $\sigma_i = \text{Softplus}(g(x, w)_{2K+i})$. Predicting the parameters of this distribution is called

Mixture Density Networks (MDN) (Bishop, 1994), which are interesting as they can represent multiple modes.

### 4.4. Minimizing the CRPS

The same concept of minimizng a SPSR can be applied for the CRPS. In fact, closed-form expressions of the CRPS are known for most canonical distributions (Jordan, 2017). For instance, for the Laplace distribution the CRPS is

$$L(z, y) = b \left( \exp\left(-b^{-1}|y - \mu|\right) - \frac{3}{4} + b^{-1}|y - \mu| \right). \qquad (13)$$

Notably, (Dheur & Taieb, 2023) compared the calibration of a MDN trained to minimize the NLL and another MDN trained to minimize the CRPS, for which a closed-form expression also exists (Grimit et al., 2006).

### 4.5. Piecewise-linear CDFs

A more flexible way to parameterize a distribution that accepts an arbitrary number of parameters is via defining the CDF at a finite number $B + 1$ of knots $(b, \tau)$. Between two consecutive knots $b_i < b_j$, no information is provided, and out of indifference, all $y : b_i < y < b_j$ are assigned the same probability density $f(y)$. This means that the CDF is composed of straight lines connecting consecutive points, justifying the piecewise linear (PL) denomination. Advantages of PL include the flexible number of parameters and the capacity to approximate any empirical distribution given enough knots. Consequently, the PDF is piecewise-constant, as we only assumed $\tau_i = F(b_i)$. Without loss of generality we can assume ordered bin borders $b_0 \leq b_1 \leq \cdots \leq b_B$.

In what follows, we provide different ways to train a network to predict the points $(b_i, \tau_i)$ that will define the CDF. We will always ask for the good simultaneous ordering of the $b_i$ and the $\tau_i$, as monotonicity ($b_i < b_j \Rightarrow \tau_i < \tau_j$) is broken otherwise, and the resulting function is not a CDF. The notion of quantile will be relevant. The $\tau$-quantile is called $b_\tau$ and it is such that $F(b_\tau) = \tau$.

### 4.6. Regression as Classification

The next loss we introduce is the categorical cross-entropy (CE) loss. The categorical cross-entropy loss is widely used for categorical distributions, i.e., where the possible set of values is unordered, discrete, finite, and mutually exclusive. These assumptions fit classification tasks, but do not fit regression. However, some notable works have used the CE loss for regression anyways (Sønderby et al., 2020; Oord et al., 2016). To compute it, they discretize the continuous domain, i.e. fix the bin borders $b_i$, and predict the probability mass ($\tau_{i+1} - \tau_i$) inside each bin $[b_i, b_{i+1}]$ for $i \in [0, B-1]$. If the bounds of the domain are unknown, one can also predict the out-of-bounds probabilities $P(Y \leq b_0)$

and $P(b_B < Y)$. The option with $(b_0, b_B)$ as borders is presented here. The Cross-Entropy loss is

$$L(z, y) = -\sum_{i=1}^{B} \mathbf{1}_{y \in [b_{i-1}, b_i]} \log z_i \qquad (14)$$

for $z_i = \text{Softmax}(g(x, w)_1, \ldots, g(x, w)_B)_i$. Due to the softmax, the predictions are normalized ($\sum_i z_i = 1$). The PDF at a given bin is given by $f(y \in [b_i, b_{i+1}]) = \frac{\tau_{i+1} - \tau_i}{b_{i+1} - b_i} = \frac{z_{i+1}}{b_{i+1} - b_i}$. Therefore, in the case of regularly spaced bins, the CE loss is equivalent to LogScore minimization up to a scale factor, and can be dubbed histogram estimation. For the case with irregular bins, the LogScore minimizing rule is

$$L(z, y) = -\sum_{i=1}^{B} \mathbf{1}_{y \in [b_{i-1}, b_i]} \log \frac{z_i}{b_i - b_{i-1}}. \qquad (15)$$

A potential drawback of this approach is that quantizing the domain and defining lower and upper bounds might not be easy. Quantile regression (explained next) might be preferred for such cases. Furthermore, this loss is local and does not exploit the well-defined bin order, which might be useful if one expects consecutive bins to have similar probabilities. Nevertheless, Stewart et al. explored why framing regression as classification was successful in machine learning competitions, and found that the implicit biases induced by gradient-descent made optimization easier for the CE loss. The observation extends trivially to probabilistic regression.

### 4.7. Quantile Regression

Quantile regression (QR) is a traditional probabilistic regression method whose goal is to predict the $b_{\tau_i}$ for a range of $\tau_i$s, for instance $[0.25, 0.5, 0.75]$. This is, the $F(b_i)$ are fixed (usually $F(b_i) = i/B$) and one predicts the $z_i = b_i$. It leverages the pinball loss, which generalizes the mean absolute error ($\tau_{\text{MAE}} = 0.5$) for any value of $\tau$. The pinball loss is

$$L(z, y) = \sum_i (\tau_i - \mathbf{1}_{y \le b_{\tau_i}})(y - b_{\tau_i}) \qquad (16)$$

and it approximates the CRPS (Bracher et al., 2021). An advantage of this loss is that each $b_{\tau_i}$ is independent, and no ordering needs to be imposed during training. This loss has been tried in reinforcement learning with success, where it was presented as an improvement to categorical distribution modeling (Dabney et al., 2018b).

### 4.8. Implicit Quantile Networks

Instead of simply doing quantile regression, one can train a network to predict the $b_\tau$ for *any* given $\tau$. This method was introduced in (Dabney et al., 2018a). Assume that the network $h : \mathcal{X} \to \mathrm{R}^d$ maps the input $x$ to some features $h(x)$, and some extra layers $g_l : \mathrm{R}^d \to \mathrm{R}$ map the features to an output value. An additional function $\phi : [0, 1] \to \mathrm{R}^d$ embeds sampled quantiles $\tau$ into the feature space. Then the estimated value is

$$b_\tau = g_l(h(x) \cdot \phi(\tau)) = g(x, w, \tau) \qquad (17)$$

with $\phi_j(\tau) = \text{ReLU}\left(\sum_{i=0}^{n-1} \cos(\pi i \tau) w_{ij} + c_j\right)$, where $n = 64$, and $w_{ij}$ and $c_j$ are the parameters of the function $\phi$. The coordinate $j$ indexes each one of the $d$ dimensions.

At each training step, one samples $N$ quantile levels $\tau_1, \ldots, \tau_N$ from a uniform distribution, predicts $z_i = b_i = g(x, w, \tau_i)$ and computes the following loss

$$L(z) = \sum_i |\tau_i - \mathbf{1}_{y \le b_{\tau_i}}| \frac{\mathcal{L}_\kappa(y - b_{\tau_i})}{\kappa}, \qquad (18)$$

where the Huber loss term (Huber, 1992) is used instead of the absolute difference of the pinball loss:

$$\mathcal{L}_\kappa(\delta) = \begin{cases} \frac{1}{2}\delta^2, & \text{if } |\delta| \le \kappa, \\ \kappa(|\delta| - \frac{\kappa}{2}), & \text{if } |\delta| > \kappa. \end{cases} \qquad (19)$$

Alternatively, one could use the pinball loss, that way an approximation of the CRPS would be minimized.

### 4.9. Energy Based Models

Introduced in (Gustafsson et al., 2020a) and further developed in (Gustafsson et al., 2022), Energy Based Models (EBM) is to the cross-entropy what IQN is to QR. However, creating an implicit version of the cross-entropy is not as easy, as the probabilities estimated by the implicit network should be normalized. In particular, EBM estimate

$$f(y) = \frac{\exp(g(x, w, y))}{Z(x, w)}, \qquad (20)$$

with a normalization term

$$Z(x, w) = \int \exp(g(x, w, y)) dy, \qquad (21)$$

which is intractable. EBM approximate the above ratio via Monte-Carlo importance sampling, therefore requiring a proposal distribution $q(y)$. The original paper assumed $q$ was a mixture of three Gaussians centered at $y$ with widths selected via hyperparameter optimization (Gustafsson et al., 2020a). It sampled $M$ values $\{b_m\}_{m=1}^M$ from $q$ and used as loss

$$L(z, y) = \log \left( \frac{1}{M} \sum_{m=1}^M \frac{\exp(g(x, w, b_m))}{q(b_m)} \right) - g(x, w, y). \qquad (22)$$

However, the choice of $q$ was arbitrary, and a workaround was proposed in (Gustafsson et al., 2022). In this second work a MDN implemented $q$ and was jointly trained with $g$ via minimization of the KL divergence with the EBM distribution, which simplified to Equation 22 (where $q$ was the MDN). This second paper also trained $g$ using noise contrastive estimation, that we omit for brevity. This method is more general and more complex than the previous ones. Similarly to IQN, EBM has a feature extractor $h$, an embedder $\phi$, and a projector $g_l$ (with different dimensions for these).

## 5. A Taxonomy of Probabilistic Regression Losses

We now develop the taxonomy of losses $L_f$, looking at (a) the distribution $f$, (b) the SPSR to be minimized, (c) the choice of the fixed, learned and predicted parameters $z$, and (d) whether the distribution is provided implicitly or explicitly. These design choices allow one to trivially generate new losses, as we exemplify by the end of this Section.

### 5.1. Distribution $f$

All the losses $L_f$ assume a probability distribution $f$ parameterized by $z$. It is impossible not to do so, as the continuous target space requires specifying probabilities for an infinite number of outcomes. The assumption on $f$ is also needed when getting distributions from ensemble methods.

There are two main groups of distributions $f$ (or $F$): canonical distributions and piecewise-linear (PL) CDFs (histogram-like PDFs). The first group uses well-known distributions that depend on a few parameters, e.g. Laplace or Mixture of Gaussians distributions (Bishop & Nasrabadi, 2006; Bishop, 1994; Hamilton et al., 2020; Dheur & Taieb, 2023). Two advantages of parametric distribution are the parameter efficiency and their natural occurrence in some applications. They should be the best if they match the underlying data-generating process. The other group assumes PL CDFs and is mainly represented by Quantile Regression (Koenker & Bassett, 1978) and Regression as Classification (Stewart et al., 2023). These are light on assumptions and flexible. The IQN (Dabney et al., 2018a) can be seen as an implicit version of QR, but EBMs (Gustafsson et al., 2022) are more general and assume little.

### 5.2. Optimization Objective

Once a distribution is chosen, it must be fitted by minimizing a criterion that encourages fidelity to the data distribution. Naturally, the two SPSR used to evaluate probabilistic forecasts are candidate criteria. The LogScore or NLL is usually well-defined and differentiable, but the CRPS is not necessarily so. Differentiable, closed-form expressions

of the CRPS for parametric distributions do exist in many cases, but exact CRPS expressions are hard to find for PL approaches. We found three differentiable expressions of the CRPS for PL CDFs that we present in the Supplement.

From the methods analyzed, most minimize the LogScore (Hamilton et al., 2020; Bishop, 1994; Gustafsson et al., 2020a; 2022; Nix & Weigend, 1994; Sønderby et al., 2020; Oord et al., 2016), but some minimize the CRPS (Dheur & Taieb, 2023; Koenker & Bassett, 1978; Dabney et al., 2018a).

### 5.3. Choice of Fixed, Learned and Predicted Parameters

Each parameter required by the distribution can be either fixed, a learned constant, or freely predicted. For instance, traditional regression using the MSE or the MAE as a loss can be seen as estimating the center of a distribution with fixed width (Bishop & Nasrabadi, 2006). However, it is also possible to learn the parameters of the distribution globally (Hamilton et al., 2020).

In particular, when assuming a PL CDF with knots $(b_i, \tau_i)$ indexed by $i$, one can choose to fix the $b_i$ and predict the free $\tau_i$ (histogram estimation) (Stewart et al., 2023) or fix the $\tau_i$ and predict the free $b_i$ (quantile regression) (Koenker & Bassett, 1978). These are the two main ways used to generate PL CDFs. Histogram estimation is easy to implement and works well given a good quantization of the domain, while quantile regression finds a good quantization of the space automatically.

### 5.4. Explicit and Implicit Models

Implicit methods implement a function $g(x, w, t)$ that describes the distribution $f$: for instance if $t = \tau$ then $g$ might predict $b = g(x, w, t)$ such that $P(Y \le b|x) = \tau$ (Dabney et al., 2018a), or conversely, $\tau = g(x, w, t = b) = P(Y \le b|x)$ (Gustafsson et al., 2020a; 2022). These examples correspond to the IQN and EBM, respectively. Implicit methods operate with fewer assumptions but are more complex to train and infer with compared to explicit methods, which are limited by the number of parameters.

### 5.5. New Losses

These design choices (distribution, objective, parameters), described in Table 1, characterize most loss functions present in previous works and signal the existence of new ones. Playing with different combinations one can get, for example, **CRPS-minimizing histogram-estimation (Hist-CRPS)**, which fixes $b_i$ and predicts $\tau_i$, but minimizing the CRPS. Another option is **Kernel Density Estimation (KDE)**, which assumes as width the variance of the predicted values and generates the centers of equally weighted Gaussians (essentially a MDN where only $\{\mu_i\}_i$ are pre-

*Table 2.* Comparison of methods across UCI datasets with mean NLL and standard deviation computed over 20-fold cross-validation (Mean ± Std). Lower is better. Colors aid visualization and go linearly from red (worse, clipped at the third maximum value) to green (better, clipped at the minimum value).

| METHOD | BOSTON | CONCRETE | ENERGY | KIN8NM | NAVAL | POWER | WINE | YACHT |
|---|---|---|---|---|---|---|---|---|
| GCDS | 18.66±8.92 | 13.64±6.88 | 1.46±0.72 | −0.38±0.36 | −5.06±0.48 | 2.83±0.06 | 6.52±21.86 | 0.61±0.34 |
| PBP | 2.53±0.27 | 3.19±0.05 | 2.05±0.05 | −0.83±0.02 | −3.97±0.10 | 2.92±0.02 | 1.03±0.03 | 1.58±0.08 |
| MC DROPOUT | 2.46±0.12 | 3.21±0.18 | 1.50±0.11 | −1.14±0.05 | −4.45±0.38 | 2.90±0.03 | 0.93±0.06 | 1.73±0.22 |
| DEEPENSEMBLES | 2.35±0.16 | 2.93±0.12 | 1.40±0.27 | −1.06±0.02 | −5.94±0.10 | 2.89±0.02 | 0.96±0.06 | 1.11±0.18 |
| CARD | 2.35±0.12 | 2.96±0.09 | 1.04±0.06 | −1.32±0.02 | −7.54±0.05 | 2.82±0.02 | 0.92±0.05 | 0.90±0.08 |
| HIST-CRPS | 3.49±0.27 | 4.15±0.13 | 3.40±0.28 | 0.17±0.10 | −2.76±0.01 | 4.24±0.07 | −0.34±0.12 | 2.78±0.24 |
| KDE | 2.86±0.38 | 3.55±1.03 | 2.75±0.48 | −1.04±0.06 | −4.22±0.14 | 2.84±0.37 | 1.02±0.32 | 3.26±0.55 |
| PINBALL | 2.78±0.30 | 3.45±0.17 | 1.55±0.40 | −0.99±0.07 | −5.73±0.83 | 2.93±0.04 | 0.80±0.08 | 1.10±0.36 |
| LAPLACE-CRPS | 2.41±0.19 | 3.04±0.10 | 1.28±0.29 | −1.03±0.02 | −5.52±0.19 | 2.89±0.02 | 0.95±0.05 | 0.93±0.18 |
| CE | 2.72±0.27 | 3.38±0.05 | 0.97±0.18 | −1.10±0.02 | −5.26±0.07 | 2.70±0.18 | −0.71±0.03 | 2.07±0.13 |
| GAUSSIAN-NLL | 2.54±0.26 | 3.21±0.42 | 1.32±0.51 | −1.25±0.05 | −6.03±0.62 | 2.81±0.04 | 0.94±0.09 | 1.41±1.41 |
| LAPLACE $w_b$ | 2.50±0.25 | 3.04±0.14 | 0.80±0.23 | −1.19±0.04 | −5.61±0.3 | 2.81±0.02 | 0.93±0.07 | 1.09±0.53 |
| MDN | 2.49±0.52 | 3.08±0.46 | 1.81±1.38 | −1.26±0.02 | −5.79±0.97 | 2.73±0.38 | 0.13±0.04 | 1.09±0.22 |
| LAPLACE-NLL | 2.43±0.30 | 3.03±0.30 | 1.16±0.17 | −1.22±0.03 | −5.68±0.24 | 2.79±0.21 | 0.93±0.04 | 0.60±0.13 |

dicted). This is a method that generates samples of the distribution in a single forward pass. This loss was evaluated in the experiments. These proposals are novel, trivially generated by changing some of the design choices described by the taxonomy, and show that it is possible to use the taxonomy as a template to create new loss functions. We evaluate them in our experiments.

## 6. Experiments

Following previous works (Han et al., 2022; Gal & Ghahramani, 2016), we conduct experiments on many real datasets of the UCI ML repository. The full dataset and experimental details are provided in the Supplement. These datasets are all tabular, and the same MLP was used for all methods. We run a subset of the methods that allow for insightful comparisons. For reference, we also report the performances of PBP (Hernández-Lobato & Adams, 2015), Monte-Carlo Dropout (MCD) (Gal & Ghahramani, 2016; Hron et al., 2018), DeepEnsembles (Lakshminarayanan et al., 2017), GCDS (Zhou et al., 2023) and CARD (Han et al., 2022), which describe approaches that are *less efficient* (regarding costs of implementation, training and inference) than direct methods (except for PBP). We also note that we trained and evaluated direct methods with the configuration tuned for CARD or simpler. Compared with CARD, we remove gradient clipping and exponential moving averages, and set weight decay to 0.01. All experimental details are provided in the Supplement.

All the PL CDF methods use 32 bins, and MDN uses 3 Gaussians. Results are summarized in Table 2, which shows that the diffusion-based method CARD performs best, followed by canonical distributions (Laplace, MDN), and them

by deep ensembles and the cross-entropy, which achieve similar performance. We again note that CARD requires plenty of training time and the hyperparameters were tuned for it to achieve good performance. The results of Table 2 are divided into sample-predicting methods (top) and direct methods (bottom). The first, except for PBP, are expensive to train and run. In contrast, direct methods are cheap. The main takeaway from the table is that direct methods are comparable in performance to sample-predicting methods. In particular, Laplace-NLL, MDN (along with its particular case Gaussian-NLL), and Cross-Entropy present performances that many times surpass those of the CARD, the best sample-predicting method. Finally, we timed the training for CARD and direct methods, finding that the direct methods are from 10× to 100× *faster* than CARD (Supplement E).

## 7. Conclusion

Probabilistic regression is relevant to many domains. Previous literature overlooked the simple, strong baselines provided by deep probabilistic regression losses. This paper organized the losses into a taxonomy, exposed closed-form expressions of the CRPS of piecewise-linear CDFs, and experimentally evaluated representative methods. When choosing a simple loss, practitioners can pick a parametric distribution and do quantile regression or histogram estimation. On top of that, they can choose to minimize the CRPS or the LogScore, which are the strictly proper scoring rules most commonly used for evaluation. Experiments suggest that canonical distributions are strong candidates and that the cross-entropy loss performs similarly to DeepEnsembles, which are top-performers for scaled up applications (Gustafsson et al., 2020b). We hope this work will draw attention to the simplest ways to do probabilistic regression.

## Acknowledgements

Blind review.

## Impact Statement

This paper presents work whose goal is to advance the field of Machine Learning. As mentioned in the Introduction, there are many potential societal consequences of our work, none which we feel must be specifically highlighted here.

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

## A. Datasets

We follow (Gal & Ghahramani, 2016; Han et al., 2022; Lakshminarayanan et al., 2017) in using public regression datasets from the UCIML repository. They can be found in this link.

## B. Code

Code is available at `https://anonymous.4open.science/r/LossesForDeepProbabilisticRegression/`.

## C. Implementation Details

We build upon CARD code. Firstly reproducing their results (they are reproducible) and then implementing our methods on their codebase. We keep our methods as simple as possible and follow their configuration (tuned for CARD) as closely as possible. The differences are: 1. we remove exponential moving averages of the weights, 2. we add 0.01 of weight decay, 3. we remove gradient clipping. For the methods that require min and max bounds, we compute the support size $r = y_{\max}^{\text{train}} - y_{\min}^{\text{train}}$, and obtain $b_0 = y_{\min}^{\text{train}} - r/10$ and $b_B = y_{\max}^{\text{train}} + r/10$ (except for Wine where we use $r/5$). The number of Gaussians for the MDN is 3, the number of bin levels and quantile levels is set to 32. We did not tune these parameters. Our NLL computation was compared with the original computation to ensure correctness. This implied adding a scaling factor to all the NLL computed in the normalized space (all the datasets are mean-std normalized). The experiments were all run on 20-fold cross-validation as done in the original (Han et al., 2022).

## D. Categorical Evaluation

Evaluating probabilistic predictions for classification or categorical tasks involves using the LogScore (as in regression) or the Brier score (Brier, 1950), although the first is preferred for mathematical convenience (Benedetti, 2010).

## E. Time Results

The time for the training over the first fold is reported for two of the datasets, namely Concrete and Kin8nm. Full comparison against CARD was not reported because CARD takes a long time to train and original results did not include training time. The results are in Table 3. They show that the cross-entropy is the slowest method, but it takes 10% of CARD training time for Concrete, and 5% of the training time for Kin8nm. The fastest of the methods in the literature is the Gaussian-NLL, which is more than $100\times$ faster than CARD.

*Table 3.* Training time in minutes across datasets and methods.

| METHOD | CONCRETE | KIN8NM |
|---|---|---|
| CARD | 9.4523 | 54.4446 |
| HIST-CRPS | 0.0906 | 0.2737 |
| KDE | 0.1446 | 0.5342 |
| PINBALL | 0.2637 | 0.4574 |
| LAPLACE-CRPS | 0.2409 | 0.6950 |
| CE | 0.9776 | 2.6564 |
| GAUSSIAN-NLL | 0.0977 | 0.4664 |
| LAPLACE $w_b$ | 0.1356 | 0.6175 |
| MDN | 0.2390 | 1.1582 |
| LAPLACE-NLL | 0.1984 | 1.5194 |

# F. Differentiable Forms of the CRPS

## F.1. CRPS Formula for PL CDF:

The consensus way to evaluate probabilistic regression predictions in some areas (e.g. meteorology, epidemiology) is using the CRPS. One can naturally ask if directly minimizing the CRPS is possible. It turns out that for the piecewise-linear CDF, the CRPS has a differentiable closed-form expression. Define borders such that $F(b_0) = 0$ and $F(b_B) = 1$. The most complicated part of the closed-form solution is the integral of $F(y)^2$ in a bin

$$\int_{b_i}^{b_{i+1}} \left( F(b_i) + \frac{F(b_{i+1}) - F(b_i)}{b_{i+1} - b_i}(z - b_i) \right)^2 dz$$
$$= -\frac{1}{3} \left( F(b_i)^2 + F(b_i)F(b_{i+1}) + F(b_{i+1})^2 \right) (b_i - b_{i+1}), \tag{23}$$

where we used that $F(y)$ is piecewise-linear. The integral of $F(y)$ in a bin is easier to derive, and we omit it for brevity. To develop the CRPS from Eq. (2) we assume $b_0 < y < b_B$, name $k$ the index such that $b_{k-1} < y < b_k$ and get $F(y)$ from linear interpolation between $F(b_{k-1})$ and $F(b_k)$. We decompose the CRPS as

$$\text{CRPS}(F, y) = \int_{-\infty}^{\infty} (F(y') - 1_{y \leq y'})^2 dy'$$
$$= \int_{b_0}^{y} F(y')^2 dy' + \int_{y}^{b_B} \left( F(y') - 1 \right)^2 dy'$$
$$= \int_{b_0}^{b_B} F(y')^2 dy' - 2 \int_{y}^{b_B} F(y')dy' + b_B - y$$
$$= b_B - y + \sum_{i=1}^{i=B} \int_{b_{i-1}}^{b_i} F(y')^2 dy'$$
$$- 2 \left( \int_{y}^{b_k} F(y')dy' + \sum_{i=k+1}^{i=B} \int_{b_{i-1}}^{b_i} F(y')dy' \right), \tag{24}$$

and find that the sum of individual applications of Eq. (23) added to the integral of $F(y)$ yields the score. For $b_B < y$

the formula is $\text{CRPS}(F, y) = \int_{b_0}^{y} F(y')^2 dy' = (y - b_B) + \int_{b_0}^{b_B} F(y')^2 dy'$ and for $y < b_0$ it is $\text{CRPS}(F, y) = (b_0 - y) + \int_{b_0}^{b_B} (F(y') - 1)^2 dy'$ (the rest follows). We refer the reader to the code for the full implementation.

This formula is used for Hist-CRPS, but it extends to all losses that use a PL CDF. We checked this implementation of the CRPS against numerical integration to ensure correctness. However, it is restricted to correctly ordered bin borders.

## F.2. Pinball loss

The pinball loss has been shown to be an approximation of the CRPS (Bracher et al., 2021). Formally, the Pinball function is

$$\rho_\tau(u) = \begin{cases} \tau\, u, & u \geq 0, \\ (\tau - 1)\, u, & u < 0. \end{cases} \tag{25}$$

and the CRPS becomes

$$\int_{-\infty}^{\infty} \left[ F(x) - \mathbf{1}\{x \geq y\} \right]^2 dx = \int_0^1 \rho_\tau\left( F^{-1}(\tau) - y \right) d\tau. \tag{26}$$

In other words, the squared-difference form of CRPS is equivalent to the integral of pinball losses over all quantiles from 0 to 1.

## F.3. Empirical CRPS for Point Predictions

Similarly to our loss KDE, one could predict many values of $\hat{y}$. Another popular form of the CRPS is

$$\text{CRPS}(F, y) = \mathbb{E}\big[|X - y|\big] - \tfrac{1}{2}\mathbb{E}\big[|X - X'|\big], \tag{27}$$

and with $M$ predictions $\hat{y}_i$ one can approximate

$$\text{CRPS}(F, y) = \frac{1}{M}\sum_i |\hat{y}_i - y| - \frac{1}{2M}\sum_i\sum_j |\hat{y}_i - \hat{y}_j| \tag{28}$$

which is differentiable with respect to the point predictions $\hat{y}$.

