# OpenReview forum: "Losses for Deep Probabilistic Regression"
_ICML.cc/2025/Conference — Submitted to ICML 2025_

### Official Review · Reviewer_xQCC · 2025-02-27

**Overall Recommendation:** 1

**Summary:**

The paper claims to be guided by the question: "What is the best probabilistic regression method?". In particular, it focuses on "direct" methods which turn supervised learning into probabilistic regression by using a different loss function. The authors summarize their contributions as introducing a taxonomy of "direct" methods, comparing them empirically to non direct methods, and providing descriptions of main concepts and evaluation practices in probabilistic regression.

---
## Update after rebuttal
The authors did not provide a rebuttal response.

**Claims And Evidence:**

(a) "the collection and categorisation of direct methods under a unifying taxonomy
- The proposed taxonomy is described in Section 5 and considers the type of distribution, optimization objective, parameters and predictions, and explicit- / implicitness
- Table 1 characterizes certain direct methods under the proposed taxonomy with columns "Minimizes", "Implicit", "Predicts", "CDF"

While I endorse the desire to introduce a unifying taxonomy, I am struggling to understand the actual unification and value provided by the proposed taxonomy. For example, in Table 1, the columns "Predicts" and "CDF", referring to type of distribution and parameters / predictions if I understand correctly, are basically still different for each method / row in the table and not unifying at all. The column "Minimizes" distinguishes between negative log-likelihood (NLL) and (approximate) continuous ranked probability score (CRPS), which I acknowledge as a meaningful differentiation. However, observing that there are mainly these two optimization objectives for probabilistic regression is not a significant contribution in my opinion. Finally, the distinction between "explicit" and "implicit" methods is explained poorly in a short paragraph 5.4. For example, from the sentence "Implicit methods operate with fewer assumptions but are more complex to train and infer with compared to explicit methods, which are limited by the number of parameters." the difference between the two is not clear to me. I believe the paragraph also contains a typographical error (because it defines implicit methods twice), making it even more confusing.

(b) "the experimental comparison against non direct methods
- Experiments are discussed in Section 6 and results are listed in Table 2, containing 14 methods evaluated on 8 datasets

Although a variety of 14 methods is great, the evaluation only considers two-layer MLPs (according to the config files in the source code) and small UCI regression datasets. In my opinion, this does not adequately address the question "What is the best probabilistic regression method?" stated in Section 1.

(c) "provide an entry-point describing the main concepts and standard evaluation practices"
- Sections 2, 3, and 4 are all a form of existing work discussion, background or review

These sections are useful and, in my opinion, the main contribution of this paper (also simply by the amount of occupied number of pages).  However, some of the discussed topics are arguably quite basic and could be found in standard machine learning / statistics textbooks. For example, Eq. (8), (9), and (10) are all dedicated towards explaining the Gaussian log-likelihood.

**Essential References Not Discussed:**

Since the paper also mentions Bayesian methods, the following [1] related review article (and all methods and articles which are discussed and cited by it) could be considered relevant.

[1] V. Fortuin. "Priors in Bayesian Deep Learning: A Review". International Statistical Review (2022), 90, 3, 563–591.

**Experimental Designs Or Analyses:**

I compliment the authors for conducting experiments over 20-fold cross-validation splits and also providing the source code. There also doesn't seem to be any issues with the soundness / validity of the experimental design itself. However, all experiments are conducted with small MLPs on small UCI regression datasets, which does not provide any evidence about how these methods would perform with other deep learning architectures and / or larger datasets.

The main takeaway seems to be that "direct" methods are competitive in performance with generative diffusion models such as CARD but much cheaper / faster. This comparison seems ill-posed to me because CARD shows that a conditional diffusion model which is pre-trained on $\mathcal{D}$ can accurately infer the predictive distribution $p(\mathbf{y} | \mathbf{x}, \mathcal{D})$ without explicitly optimizing evaluation metrics, such as MSE or negative log-likelihood, whereas the sole purpose of "direct" methods is to directly optimize such metrics in a supervised learning setting.

**Methods And Evaluation Criteria:**

The paper suggests two new losses based on the proposed taxonomy (Hist-CRPS and KDE). However, the empirical evaluation of the paper itself shows that these are among the worst methods in terms of predictive NLL. Otherwise, the paper does not propose any methods, given that it is mainly a review paper.

The benchmark datasets are from the popular UCI regression benchmark. They are small yet popular in research communities such as Bayesian deep learning. I personally think it is time to move on and find new problems / benchmarks in this field, but this may be more general criticism which is not directly related to this particular paper.

**Other Comments Or Suggestions:**

- Author names of narrative in-text citations should not be surrounded by parentheses
- Paragraph 5.4. "Explicit and Implicit Models" describes implicit methods twice. Is this perhaps a typographical mistake or intended?

**Other Strengths And Weaknesses:**

Strengths:
- detailed descriptions which would be accessible to someone who is entirely new to the field
- comprehensive discussion on the motivation for probabilistic regression with corresponding citations

Weaknesses:
- the introduced taxonomy does not seem to be very unifying except for distinguishing between CRPS and NLL objectives
- some discussed topics seem quite basic and are also available in standard textbooks or on Wikipedia

**Questions For Authors:**

1. What is the difference between "explicit" and "implicit" methods, proposed as part of the taxonomy introduced in the paper?

**Relation To Broader Scientific Literature:**

The paper is primarily a review paper which discusses a broader variety of approaches rather than contribution a particular method.

**Theoretical Claims:**

The main paper does not make any substantial theoretical claims. The appendix includes some calculations for differentiable forms of the CRPS, which I did not check with great detail.

---

### Official Review · Reviewer_LF5Q · 2025-03-11

**Overall Recommendation:** 1

**Summary:**

The paper discusses losses for deep probabilistic regression. The authors identify the gap in the literature of scattered knowledge on deep probabilistic regression across various domains and propose a taxonomy of the method to unify the knowledge in that area. Based on that, they identify easy-to-achieve new deep probabilistic regression losses and methods. Moreover, the authors perform an experimental evaluation of the various discussed "direct" methods with "non-direct" ones.

**Claims And Evidence:**

The papers provide three main claims:
(a) the collection and categorization of direct methods under a unifying taxonomy;
(b) the experimental comparison against non direct methods;
(c) provide an entry-point describing the main concepts and standard evaluation practices.

The overall evidence for the claims is weak.

For (a), Section 5 describes the taxonomy, but it's not the easiest to follow and lacks clarity. For example, in Section 5.1, why aren't more probabilistic distributions discussed? Or what are the explicit models in Section 5.4? I would expect the taxonomy to have proper definitions and clarity. Finally, some form of visualization of the taxonomy would be helpful.

For (b), please see the following sections on methods, evaluation criteria and experimental design.

For (c), Sections 3 and 4 contain the background and mentioned entry to the domain. In Section 3, it's not clear why the sharpness, calibration, reliability diagram, and Expected Calibration Error are discussed. Additionally, I lack good formalism with clear descriptions of the variables, parameters, and so on.

**Essential References Not Discussed:**

No.

**Experimental Designs Or Analyses:**

I evaluated the soundness and validity of the experimental part of the work presented in Section 6. I lack the more rigorous and extensive evaluation of the methods.

First of all, "Results are summarized in Table 2, which shows that the diffusion-based method CARD performs best" - based on what is this conclusion? CARD method is not consistently the best one, e.g., Energy dataset, Power dataset, Wine dataset, Yacht dataset. There is no aggregated summary of the performance of the methods.

Second of all, I miss additional analysis that would discuss the inference times, sample size analysis, and training times (in more detail).

Third of all, the paper discusses the probabilistic regression - I miss at least one or two examples of the estimated distributions based on various methods and discussion on them.

Question: Why don't authors evaluate methods using CRPS?

Question: What is the architecture of the used MLP network? What is the number of parameters? What is the impact of MLP network size on the results?

Question: What infrastructure was used for the experiments?

Suggestion: The author could consider the experiment on an artificial dataset created using known probabilistic distribution and check the methods' effectiveness. Check [5] for a further reference.

**Methods And Evaluation Criteria:**

The proposed methods don't seem to be exhaustive, and the presentation form doesn't make it easy to follow the logic of selection. Namely, the authors propose a taxonomy of the methods, and I would expect to see some cartesian product in the proposed methods, i.e., {distribution1, ..., distribuionN} x {loss1, loss2} x {...}. Also, the methods are limited to only Gaussian, Laplace, and Mixture of Gaussian parameteric distributions, and it's not clear why other distributions are not evaluated, e.g., t-Student, Logistic, LogNormal, Gumbel, Weibull, Poisson, or Negative Binomial (per [1]). Additionally, I miss the comparison with normalizing flow methods like NICE [2], RealNVP [3], or MAF [4] and the follow-up methods dedicated to probabilistic regression like TreeFlow [5] or NodeFlow [6]. Finally, even though the paper focuses strictly on the deep models, it would be fair to compare them with tree-based methods like CatBoost [7] or NGBoost [8].

The proposed evaluation criteria, to some point, make sense, but they are not exhaustive enough. While the benchmark from Gal & Ghahramani (2016) is a de facto standard, I would expect the "review paper" (as the paper positions itself) to make a more comprehensive comparison based on an OpenML benchmark like one in the TabPFN article [9].

[1] "Probabilistic Gradient Boosting Machines for Large-Scale Probabilistic Regression", Sprangers et al., 2021
[2] "NICE: Non-linear Independent Components Estimation", Dinh et al., 2014
[3] "Density estimation using Real NVP", Dinh et al., 2016
[4] "Masked Autoregressive Flow for Density Estimation", Papamakarios et al., 2017
[5] "TreeFlow: Going beyond Tree-based Gaussian Probabilistic Regression", Wielopolski & Zieba, 2022
[6] "NodeFlow: Towards End-to-End Flexible Probabilistic Regression on Tabular Data", Wielopolski et al., 2024
[7] "Uncertainty in Gradient Boosting via Ensembles", Malinin et al., 2020
[8] "NGBoost: Natural Gradient Boosting for Probabilistic Prediction", Duan et al., 2020
[9] "TabPFN: A Transformer That Solves Small Tabular Classification Problems in a Second", Hollmann et al., 2022

**Other Comments Or Suggestions:**

- 019, Right: "Methods that mirror supervised learning,"
	- What do you mean by mirror?
- 020, Right: "are particularly attractive when considering efficiency, ease of use and scalability. "
	- Why?
- 058, Left: "sampling methods and direct methods"
	- What do authors mean by sampling methods? They are not discussed or explained previously.
- 146, Left: "R"- styling: the convention for real numbers is $\mathbf{R}$.
- 150, Right: Citation of SPSR could be earlier in 126, Right, as it's the first time when it is mentioned.
- 416, Left - "real datasets" - "real-world datasetes"?
- 435, Right - "canomical" - typo
- Appendix, Training Time, Table 3, Readability of the results could be improved, for example it would be better to have minutes + seconds?

**Other Strengths And Weaknesses:**

- 155, Left - The paper is not comprehensive to all deep probabilistic regression but limited to univariate ones - it's worth mentioning in the beginning. Moreover modeling $p(y_1|x)$, ..., $p(y_N|x)$ is a special case of modeling $p(y_1, ..., y_n|x)$ - it's might be also worth mentioning.

**Questions For Authors:**

Questions in the previous sections.

**Relation To Broader Scientific Literature:**

See the methods and evaluation criteria sections.

The additional question to that is why authors don't include the analysis of multivariate probabilistic regression.

**Theoretical Claims:**

No theoretical claims.

---

### Official Review · Reviewer_s37i · 2025-03-14

**Overall Recommendation:** 2

**Summary:**

The authors began their research by observing that despite using probabilistic regression across various fields, there was no unified overview of these methods. First, they analyze various Probabilistic regression approaches and organize them from the perspective of "closed-form expressions of the CRPS of piecewise-linear CDFs." Based on this, they organize representative methods and conduct comparative experiments on UCI datasets. Through these experiments, they demonstrate that direct methods are not only simpler to train and less costly to infer compared to sample prediction methods but also achieve similar performance. Therefore, the authors encourage the community to reconsider the effectiveness of basic methods.

**Claims And Evidence:**

* The authors' claims are adequately supported through comparative experiments. However, despite using multiple regression datasets from UCI, additional validation using different datasets would further enhance the reliability of their findings.

**Essential References Not Discussed:**

* To my knowledge, this paper adequately covers relevant works in the field.

**Experimental Designs Or Analyses:**

* The experiments designed to compare probabilistic regression methods were derived from the experimental design derived from this basis [Han et al., 2022; Gal & Ghahramani, 2016]. This makes it technically valid since they compared it with CARD. They also shared their environment settings through links in the supplementary materials.

**Methods And Evaluation Criteria:**

* The authors aim to analyze deep regression loss from multiple angles. To this end, they integrate the theoretical foundations of various losses used in different papers and verify through repeated experiments whether these losses can easily achieve good performance through the actual learning process. Additionally, they designed experiments by selecting the commonly used UCI dataset in deep regression loss research.

**Other Comments Or Suggestions:**

* In Table 2, I recommend modifying the table to better highlight the losses designed with the design choices described in section 5.5.
* For the completeness of the paper, I recommend adding experimental environment details to the supplementary materials, rather than just a link.

**Other Strengths And Weaknesses:**

* This paper classifies various losses in Probabilistic regression and clearly conveys their characteristics through repeated comparative experiments, providing other researchers with a powerful and simple baseline. As the authors note, the wide variety of losses used across different application fields makes meaningful comparisons challenging.  Therefore, as a researcher, I highly welcome this study that establishes such a baseline.

* It is interesting that they compared not only average performance across various datasets but also hyperparameter settings and actual computation time to evaluate the priority of loss functions.

* The nature of this paper differs from typical conference papers. That is, it is difficult to evaluate it based on criteria such as the novelty of a method.

**Questions For Authors:**

The losses generated through the design space proposed by the authors appear to perform worse compared to other methods. From this perspective, please explain whether considering the design space is meaningful and why these losses showed degraded performance.

**Relation To Broader Scientific Literature:**

* Given that Deep Probabilistic Regression is used across various fields, this research will help other scientific studies produce more reliable prediction results.

**Theoretical Claims:**

* This paper does not make any theoretical claims. Their main claims are experimentally demonstrated.
* Their process of organizing various forms of loss into the proposed taxonomy is very reasonable.

---

### Official Review · Reviewer_WYfL · 2025-03-14

**Overall Recommendation:** 1

**Summary:**

The paper offers a summary of probabilistic regression methods, primarily focusing on so-called "direct methods" in the supervised regime. The review discusses at length the strictly proper scoring rules: Continuous Ranked Probability Score and the Negative Loglikelihood. Various probabilistic methods are discussed along with losses. The paper provides experimental comparisons of mean NLL values over 8 datasets spanning a variety of probabilistic regression models.

**Claims And Evidence:**

The main claim of this review paper is the development of a taxonomy of direct methods. While there is an attempt at this, it is not done so very clearly or concisely thus failing as a taxonomy.

**Essential References Not Discussed:**

N/A

**Experimental Designs Or Analyses:**

The evaluation of NLL over a span of models and datasets is overall rather uninformative. The choice of method still depends on domain information and as such there is no "best" probabilistic regression method. The nuances of this are not discussed.

**Methods And Evaluation Criteria:**

N/A

**Other Comments Or Suggestions:**

N/A

**Other Strengths And Weaknesses:**

Since this paper is a survey paper, it is overall not particularly original. The significance of it is also difficult to pin-point since practitioners would most likely be interested in domain specific surveys/reviews of the topic. A survey should either act as a clear introduction to a topic allowing for further investigation in open problems or a consolidation of related research in a clear and concise manner. This paper fails to do either of these tasks as it seems confused for what or for whom it is applicable over the existing cited surveys. Additionally, for the breadth of topics covered, I do not think that the length of a conference paper is sufficient to truly cover the intended information in the clear and well developed manner of a survey.

**Questions For Authors:**

N/A

**Relation To Broader Scientific Literature:**

This serves as a summary of losses for a large breadth of direct probabilistic regression methods. It does not necessarily improve on the domain specific papers cited.

**Theoretical Claims:**

N/A

---

### Decision · Program_Chairs · 2025-05-01

**Decision:**

Reject

**Comment:**

The reviewers agree that while the paper has a valuable motivation, it falls short in execution. They find the proposed taxonomy unclear and not meaningfully unifying, the experiments limited and uninformative, and the new methods underperforming. Key classes of related work are missing, and the writing lacks clarity. With no author engagement during the rebuttal phase, the reviewers collectively recommend rejection due to insufficient clarity, depth, and contribution.